# Dual-Toxin-Producing *Clostridium botulinum* Strain Isolated from a Foodborne Botulism Case in Korea: Genomic and Functional Insights

**DOI:** 10.3390/toxins17060299

**Published:** 2025-06-12

**Authors:** Eun-Sun Choi, Chi-Hwan Choi, Jun-Ho Jeon, So-Hyeon Kim, Hyun-Ju Song, Hwajung Yi, Gi-eun Rhie, Yoon-Seok Chung

**Affiliations:** 1Division of High-Risk Pathogens, Department of Laboratory Diagnosis and Analysis, Korea Disease Control and Prevention Agency, KDCA, Cheongju 28159, Republic of Korea; 2Division of Laboratory Diagnosis and Analysis, Chungcheong Regional Center for Disease Control and Prevention, Korea Disease Control and Prevention Agency, KDCA, Daejeon 35208, Republic of Korea

**Keywords:** *Clostridium botulinum*, botulinum toxins, dual-toxin-producing strain, bont/b, bont/f, foodborne botulism

## Abstract

*Clostridium botulinum* produces one of the most potent biological toxins and causes botulism, a rare but potentially fatal neuroparalytic disease. In 2014, a foodborne botulism case was reported in Korea, and a strain (CB-2014001) was isolated. Initial characterization identified it as a BoNT/B-producing strain based on mouse bioassay and conventional PCR. However, subsequent genomic analysis revealed the presence of dual BoNT gene clusters, bont/B and bont/F, corresponding to subtypes B5 and F2, respectively. Therefore, we aimed to analyze the genetic characteristics and toxin expression profiles of the isolated strain. The strain showed high sequence identity with Bf-type strains such as CDC 3281 and An436. Functional assays confirmed simultaneous expression of both BoNT/B and /F toxins at 35 °C, and temperature-dependent assays revealed predominant expression of BoNT/F at 30 °C and BoNT/B at 37 °C, indicating that toxin expression is influenced by environmental temperature. These findings highlight the potential for differential pathogenicity based on culture conditions and underscore the importance of developing diagnostic tools capable of detecting multiple *bont* genes. To our knowledge, this is the first report of a dual-toxin-producing *C. botulinum* strain associated with foodborne botulism in Korea, providing important insights into botulism diagnosis, treatment strategies, and public health preparedness.

## 1. Introduction

*Clostridium botulinum* is a gram-positive, obligate anaerobic, and endospore-forming bacterium that produces botulinum neurotoxins (BoNTs). Serologically, BoNTs are classified into seven types, A through G, with types A, B, E, and F causing most cases of botulism in humans [1]. From 2008 to 2019, an average of 170 botulism cases per year were laboratory-confirmed in the United States, including infant botulism (74%), wound botulism (13.4%), foodborne botulism (11.1%), and cases of unknown etiology (1.5%). Among foodborne botulism cases reported during this period, the detected botulinum toxin types were as follows: A (70%), B (6.8%), E (21.4%), F (1.8%), and unknown (1.4%) [2]. In Korea, from 2001 to 2024, a total of 11 botulism cases were reported, including one case of infant botulism, eight cases of foodborne botulism, and two cases of unknown etiology [3]. The botulinum toxin types detected in these cases were A (three cases), B (five cases), Ab (one case), and E (one case). In 2019, the first case of infant botulism was reported in Korea. The *C. botulinum* strain (CB-27) isolated from the patient was genetically characterized to harbor both *bont/A* and *bont/B* toxin genes; however, only *bont/A* was functionally expressed, while *bont/B* remained silent. This marked the first identification in Korea of a strain possessing dual-toxin gene clusters [4].

In 2014, clinical specimens (stool and serum) collected from a patient suspected of having foodborne botulism in Korea were submitted to the Korea Disease Control and Prevention Agency (KDCA) for laboratory diagnosis. Analysis of the stool sample using a mouse bioassay and monovalent antitoxin B (National Institute for Biological Standards and Control, Potters Bar, UK) confirmed the presence of botulinum neurotoxin B (BoNT/B), and the case was diagnosed as foodborne botulism caused by BoNT/B. However, no official case report detailing the clinical course or treatment has been published to date. *C. botulinum* was subsequently isolated from the stool sample and designated as strain CB-2014001. However, a later genomic analysis in 2020 revealed that this strain harbors both *bont/B* and *bont/F* gene clusters. Apart from single-type toxins, dual-type toxins (Ab, Ba, Bf, and Af) produced by *C. botulinum* strains have rarely been reported [5,6,7,8,9]. According to recent botulism surveillance data from the United States Centers for Disease Control and Prevention (CDC), dual-toxin-producing strains such as Bf, Ba, and Af have primarily been isolated from clinical specimens in infant botulism cases, whereas no such strains have been reported in foodborne botulism [2]. Foodborne botulism involving Bf-type strains has only been documented in the United States in 1997 [10,11] and in France in 2008 [12]. The identification of *bont/B* and *bont/F* dual-toxin gene clusters in a foodborne *C. botulinum* strain constituted the first such finding in Korea and highlighted a rare event that has only occasionally been reported on a global scale. This raised important questions about the functional expression of both toxins and the strain’s classification within the BoNT subtype framework.

Therefore, this study aimed to determine whether strain CB-2014001 expresses and secretes both BoNT/B and BoNT/F, and to genetically characterize the strain to clarify its identity as a dual-toxin-producing *C. botulinum*. Previous studies, such as that by Jason et al. [13], have demonstrated that temperature-dependent culture conditions can influence toxin expression in dual-toxin-producing strains. Based on this, CB-2014001 was cultured at two different temperatures, and mouse neutralization assays were performed using the culture supernatant. This study ultimately aimed to enhance understanding of botulism pathogenesis and epidemiology by elucidating the genetic and functional properties of a rare dual-toxin-producing strain isolated in Korea.

## 2. Results

### 2.1. Genomic Analysis of CB-2014001

Whole genome annotation revealed that strain CB-2014001 possesses two distinct *bont* gene clusters, both located on a single plasmid: *bont/B* (*ha17*, *ha33*, *ha70*, *botR*, *ntnh*, *bont*) and *bont/F* (*orf-X3*, *orf-X2*, *orf-X1*, *botR*, *ntnh*, *bont*) (Figure 1). The lengths of these clusters were 11,697 and 14,869 bp, respectively, and were deposited in GenBank under accession numbers OP019001 (*bont/B*) and OP019002 (*bont/F*). Comparative BLAST (version 6.4.5) analysis using 477 genomes from the GenBank database [14] showed that the *bont* genes of CB-2014001 shared high sequence similarity with known dual-toxin-producing strains, such as CDC 3281 (B: 99.97%, F: 99.87%), Bf (B: 99.48%, F: 99.90%), An436, and CDC69057. Phylogenetic analysis using the GGDC web service classified the *bont/B* and *bont/F* clusters as belonging to subtypes B5 and F2, respectively (Figure 2) [15]. Protein-level comparisons against BoNTbase confirmed their identity as the B5 (99.7–99.9%) and F2 (99.5–99.7%) subtypes [16]. Subtypes are defined based on amino acid sequence differences within the same serotype, with a threshold of more than 2.6% divergence typically used to distinguish one subtype from another [17].

### 2.2. Recultivation and Mouse Bioassay of CB-2014001

To determine whether both BoNT/B and BoNT/F are functionally expressed, CB-2014001 was TPGY broth. Based on previously reported studies [14,18], the strain was cultured at 35 °C, the optimal temperature for the production of both toxins. The culture supernatant was filtered and diluted by 1:640 with phosphate-buffered saline containing 0.2% gelatin, then mixed with 0.5 IU each of monovalent antitoxins B and F. The resulting mixture was injected intraperitoneally into ICR/CD-1 mice. Mice injected with the culture filtrate alone died, whereas those injected with heat-treated filtrate survived, indicating that the toxicity was heat-labile. Moreover, neutralization was observed only when both antitoxins B and F were used in combination, not when either was used individually (Table 1), confirming the presence of both functional BoNT/B and BoNT/F.

### 2.3. Temperature-Dependent Expression of BoNT/B and BoNT/F

To evaluate the effect of temperature on toxin expression, strain CB-2014001 was cultured in TPGY broth at 30 and 37 °C for 18 h. Each culture supernatant was diluted by 1:640 and subjected to mouse neutralization assays using monovalent antitoxins B and F. Injection of the filtrate obtained at 30 °C into ICR/CD-1 mice caused lethality, which was neutralized by antitoxin F, whereas antitoxin B alone failed to protect the mice. In contrast, mice injected with the 37 °C culture filtrate survived when treated with antitoxin B, but not with antitoxin F (Table 2). These results suggest that strain CB-2014001 exhibits differential expression of BoNT/F and BoNT/B depending on the incubation temperature.

## 3. Discussion

The genomic and functional analyses of strain CB-2014001 collectively support its classification as a dual-toxin-producing *C. botulinum* strain. Genomic annotation revealed the presence of two distinct *bont* gene clusters: *bont/B* (ha17, ha33, ha70, botR, ntnh, bont) and *bont/F* (orf-X3, orf-X2, orf-X1, botR, ntnh, bont), with each cluster showing high sequence identity to known dual-toxin-producing strains, including CDC 3281 and An436. Phylogenetic analysis and comparison with BoNTbase confirmed that the *bont/B* and *bont/F* clusters belong to the B5 and F2 subtypes, respectively. Importantly, experimental evidence demonstrated that both BoNT/B and BoNT/F were functionally expressed and secreted when strain CB-2014001 was cultured at 35 °C, an optimal condition reported in previous studies [13,19] for inducing dual-toxin production. The culture supernatant was lethal to mice but lost toxicity after heat treatment, indicating heat-labile toxin activity. Furthermore, neutralization of toxicity was achieved only by the combination of antitoxins B and F, and not by either alone, confirming the co-expression of both toxins.

In addition to confirming the presence and co-expression of both toxins, our study also revealed that toxin production by CB-2014001 is temperature dependent. Mouse neutralization assays demonstrated that BoNT/F was predominantly expressed at 30 °C, while BoNT/B was dominant at 37 °C. Based on the lethal dilution of the culture supernatants, the filtrate obtained at 37 °C was lethal to mice at a 1:2560 dilution, whereas that from the 30 °C culture was lethal only up to a 1:640 dilution, indicating approximately fourfold greater toxicity at the higher temperature (Appendix A). This suggests that strain CB-2014001 produced substantially more toxin at 37 °C, where BoNT/B was the predominant type, compared with that at 30 °C, where BoNT/F was primarily expressed. Considering that BoNT/B was significantly more abundant at 37 °C, it is reasonable to infer that BoNT/B may also be more dominantly expressed at 35 °C, although direct quantification at this temperature was not performed. These findings imply that environmental temperature influences not only the type of BoNT expressed but also the total amount of toxin produced, which may, in turn, affect disease severity in real-world infection scenarios. The temperature-dependent differences in BoNT/B and BoNT/F expression are reported not to be genetically driven [19]. The botR gene, a transcriptional regulator of the botulinum toxin gene cluster, controls the expression of bont/A and the associated ha operon. In their study, Couesnon et al. observed that the expression levels of botR/A, bont/A, and ha17 remained similar at both 37 °C and 44 °C. However, toxin activity was markedly reduced at 44 °C, which was not due to transcriptional repression but rather the degradation of BoNT/A and NTNH/A proteins by temperature-induced, calcium-dependent proteases. These findings suggest that the botR gene and promoter regions are not directly temperature-sensitive, and that post-transcriptional regulation and protein stability play a more critical role in temperature-dependent toxin expression. It is also plausible that the high-temperature degradation mechanism observed for BoNT/A may apply to BoNT/F, suggesting that BoNT/F protein may be more rapidly degraded at 37 °C, thereby contributing to its lower activity at this temperature. To build on these observations, future studies will focus on quantifying the relative expression levels of BoNT/B and BoNT/F under co-expression conditions through protein-level analyses. Understanding which toxin is more dominantly produced will be critical for optimizing type-specific antitoxin dosing strategies. This knowledge will also contribute to a more precise understanding of the pathogenic potential of dual-toxin-producing *C. botulinum* strains and support the development of more effective therapeutic approaches.

Although dual-toxin-producing *C. botulinum* strains are relatively rare, infections caused by these strains may be associated with increased clinical severity and present additional challenges for diagnosis and treatment. A previous study reported 30 cases of infant botulism in California between 1976 and 2020 that were attributed to dual-toxin-producing strains of the Ba and Bf types [20]. Among these cases, 60% of patients required intubation upon hospital admission, a significantly higher rate than that observed in patients infected with BoNT/B alone (34.3%) [20]. Notably, the strains produced BoNT/B5 and BoNT/F2 subtypes, and all toxin gene clusters were plasmid-encoded. These findings suggest that infections involving dual-toxin-producing strains may lead to more severe clinical outcomes—such as increased symptom severity and longer hospitalization—compared to infections caused by single-toxin-producing strains. In Korea, two such cases were also identified: one in 2014 and another in 2019. In the 2014 case, an Ab-type strain was identified, in which the bont/B gene was silent and the bont/A gene was plasmid-encoded. In the 2019 case, both the bont/B and bont/F genes were found to be plasmid-borne. However, the mechanisms underlying the emergence of such dual-toxin-producing strains remain unclear. Current treatment options for botulism include human botulism immune globulin (BIG-IV), which is primarily effective against BoNT/A and BoNT/B [21], and botulism antitoxin heptavalent (BAT), which provides coverage against all seven BoNT serotypes (A–G) [22]. Nonetheless, to effectively respond to cases caused by diverse or newly emerging toxin variants, it is crucial to develop broadly protective and safer antitoxin therapies with minimal side effects [23,24].

To prepare for future outbreaks caused by atypical or emerging *C. botulinum* strains—including those producing dual toxins—the Korea Disease Control and Prevention Agency (KDCA) is strengthening its public health infrastructure. In Korea, botulinum neurotoxin is not only classified as a Group 1 nationally notifiable infectious disease but is also designated as a biological threat agent. Therefore, future efforts will focus on elucidating the mechanisms behind dual-toxin expression and on advancing rapid diagnostic methods, genome-based analytical tools, and effective antitoxin treatments to enhance clinical response and reinforce public health preparedness.

## 4. Conclusions

In this study, we characterized the *C. botulinum* strain CB-2014001, a dual-toxin-producing isolate identified in Korea. Genomic and phylogenetic analyses revealed that the strain harbors two distinct *bont* gene clusters, *bont*/B and *bont*/F, corresponding to subtypes B5 and F2, respectively, and exhibits high sequence identity with previously reported Bf-type strains. Functional expression of both BoNT/B and BoNT/F was confirmed through mouse bioassays, with neutralization achieved only when antitoxins B and F were administered together. Temperature-dependent expression assays demonstrated that BoNT/F was predominantly produced at 30 °C, whereas BoNT/B was predominant at 37 °C. Quantitative analysis further revealed that BoNT/B was produced at approximately fourfold higher levels at 37 °C compared with that of BoNT/F at 30 °C, highlighting the significant influence of environmental conditions on both toxin type and total toxin production.

These findings underscore the critical importance of identifying and characterizing dual-toxin-producing *C. botulinum* strains to ensure accurate diagnosis and effective treatment. In parallel, the development of rapid diagnostic tools capable of detecting multiple *bont* genes and broad-spectrum antitoxins effective against multiple BoNT types will be essential for strengthening public health preparedness and response to future botulism outbreaks.

## 5. Materials and Methods

### 5.1. Bacterial Strain

*C. botulinum* strain CB-2014001, isolated from a patient with foodborne botulism in Korea, was used in this study. The strain was initially cultured on egg yolk agar under anaerobic conditions at 35 °C for 48 h, and then sub-cultured into trypticase-peptone-glucose-yeast extract (TPGY) broth for further experiments. The liquid culture was incubated anaerobically at 30, 35, and 37 °C for 18 h.

### 5.2. Mouse Bioassay

Mouse bioassays were conducted to assess the presence and types of BoNT. The cultured broth was centrifuged and filtered through a 0.2-µm disposable membrane filter. The filtrate was diluted 1:640 in phosphate-buffered saline containing 0.2% gelatin. Monovalent antitoxins B and F (0.5 IU each) were added to the diluted filtrate, and 0.5 mL of the mixture was intraperitoneally injected into ICR/CD-1 mice (female, 4–5 weeks old, 18–20 g). Mice were monitored for survival, and neutralization assays were performed to determine the presence of BoNT/B and BoNT/F. The animal experiments were approved by the Institutional Animal Care and Use Committee of the KDCA (grant number: KDCA-IACUC-008-20-1A;2020).

### 5.3. Genomic DNA Extraction

Genomic DNA was extracted from *C. botulinum* CB-2014001 cultured under anaerobic conditions in TPGY medium at 35 °C for 18 h. DNA was isolated using the Wizard^®^ Genomic DNA Purification Kit (Promega, Madison, WI, USA), following the manufacturer’s protocol optimized for Gram-positive bacteria. Briefly, bacterial cells were harvested by centrifugation at 8000× *g* for 5 min, resuspended in lysis buffer containing lysozyme (10 mg/mL), and incubated at 37 °C for 30 minute to ensure efficient cell wall degradation. Subsequent protein precipitation and DNA purification steps were carried out as instructed in the kit protocol. The concentration and purity of the extracted DNA were assessed using a NanoDrop spectrophotometer (Thermo Fisher Scientific, Waltham, MA, USA), and the integrity was verified by agarose gel electrophoresis.

### 5.4. Genomic Analysis

Whole-genome sequencing of strain CB-2014001 was performed using the PacBio RS II (Pacific Biosciences) and the Illumina MiSeq (Illumina) platforms. De novo genome assembly was conducted using CANU v1.7 and SPAdes v3.14.0. High-quality MiSeq reads were used to correct errors in PacBio long reads using Proovread. Genome annotation was performed using the NCBI Prokaryotic Genome Annotation Pipeline (PGAP).

## Figures and Tables

**Figure 1 toxins-17-00299-f001:**
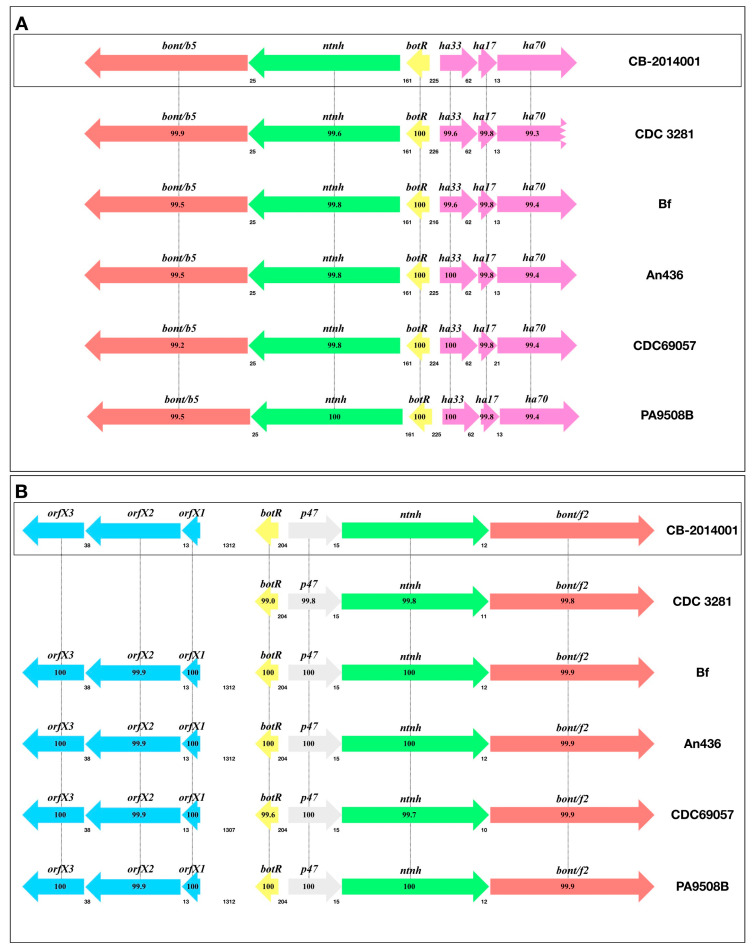
*bont* gene cluster arrangements in strain CB-2014001 and other known representative strains, including (**A**) Subtype B5; (**B**) subtype F2. Arrows indicate the gene location and orientation; gene names are shown above the arrows; the numbers between genes below the arrows indicate intergenic spacing. The percentage values below each gene represent Average Nucleotide Identity (ANI) compared to the corresponding gene in strain CB-2014001. Dotted lines connect homologous genes across strains to highlight sequence conservation patterns. GenBank accession numbers are CB-2014001 (B: OP019001, F: OP019002), CDC 3281 (B: Y13630, F: Y13631), Bf (ABDP00000000), An436 (LFON00000000), CDC69057 (QVAH00000000), and PA9508B (CP102918).

**Figure 2 toxins-17-00299-f002:**
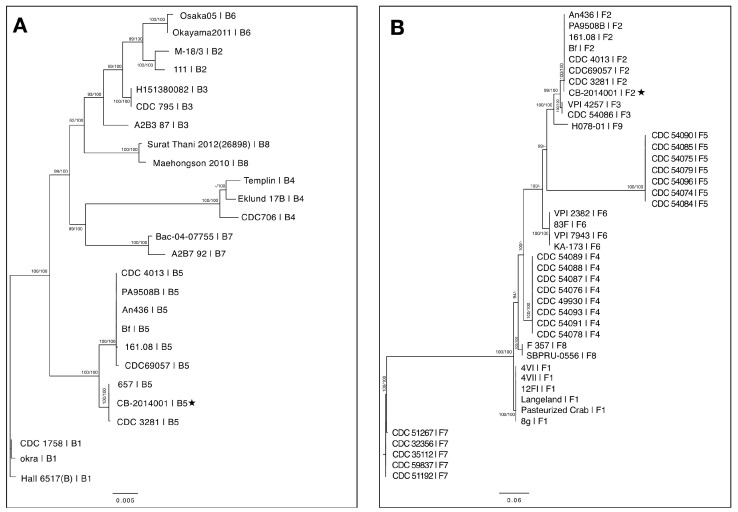
Phylogenetic tree of the *bont/B* (**A**) and the *bont/F* (**B**) nucleotide sequences. Maximum likelihood (ML) tree inferred under the GTR+GAMMA model and rooted via midpoint-rooting. The branches are scaled in terms of the expected number of substitutions per site. The numbers above the branches are support values when > 60% from the ML (**left**) and maximum parsimony (MP) (**right**) bootstrapping. Detailed information for *bont* genes is provided in Appendix A.

**Table 1 toxins-17-00299-t001:** Determination of toxin types produced by *C. botulinum* strain CB-2014001 using mouse neutralization assay.

Condition	No. of Mice Alive/Tested	Toxin Type Detected
**Untreated filtrate**	0/2	-
**Heat-treated filtrate**	2/2	-
**Filtrate + antitoxin B**	0/2	-
**Filtrate + antitoxin F**	0/2	-
**Filtrate + antitoxin B + F**	2/2	B and F

Note: The filtrate was heated for 10 min in a boiling water bath.

**Table 2 toxins-17-00299-t002:** Mouse neutralization assay results for culture filtrates of strain CB-2014001 at different temperatures.

Condition	No. of Mice Alive/Tested		Toxin Type Detected	
	30 °C	37 °C	30 °C	37 °C
**Untreated filtrate**	0/2	0/2	-	-
**Heat-treated filtrate**	2/2	2/2	-	-
**Filtrate + antitoxin B**	0/2	2/2	-	B
**Filtrate + antitoxin F**	2/2	0/2	F	-

## Data Availability

The genomic sequences of the *bont/B* and *bont/F* gene clusters from strain CB-2014001 have been deposited in the GenBank database under accession numbers OP019001 and OP019002, respectively. Additional data supporting the findings of this study are available from the corresponding author upon reasonable request.

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
