# Peer review of "Dual-Toxin-Producing Clostridium botulinum Strain Isolated from a Foodborne Botulism Case in Korea: Genomic and Functional Insights"

_toxins, 2025, doi:10.3390/toxins17060299_

Round 1
Reviewer 1 Report
Comments and Suggestions for Authors
The main point of this manuscript is the detection of genes involved in the pathogenesis (mechanism by which the disease develops) of human botulism. The topic and field of study are relevant to the pathology and toxicology areas. I consider the genes involved in the disease's mechanism an interesting drive and the possible pathological manifestation in the outbreak episodes. In my opinion, the relevant aspect of the genes or the effect of the genes in the general context of disease is a differentiating point from many other studies.
The authors should consider the detention of genes and the simplicity and details of the description of methods for specific improvements regarding the methodology.
The conclusions are consistent with the evidence and arguments presented. I'd like to point out that it's true to detect genes relative to outbreaks, but at this point, the authors were balanced in the conclusion at light of your results and perspectives. I agree with the statement: ...the critical importance of identifying and characterizing dual-toxin-producing C. botulism...diagnostic and effective treatment.
The references are appropriate and the tables and figures were succinct and pointed relative to showing the principal results of the study.
In conclusion, the manuscript under consideration (Toxins-3668807) offers a noteworthy perspective on the pathogenesis of botulism, a prevalent infection affecting the global population. The manuscript is appropriate and well-written. The abstract is concise and comprehensive, encompassing all phases of the research. The introduction exhibits an appropriate and satisfactory flow for the reader. The "Materials and Methods" section describes and elucidates the main methodological points; the results are well distributed, organized, and easily comprehensible. The discussion discourse is pertinent and focused on the objectives and results of the study. The scientific writing in English is consistent with the proposal of the Toxins journal. After careful consideration, I have decided to accept for publication.
Author Response
Response to Reviewer 1
We sincerely thank Reviewer 1 for the thoughtful and encouraging comments on our manuscript. We greatly appreciate your recognition of the scientific merit, clarity, and relevance of our study, as well as your constructive suggestions for improvement.
[Comment]: The authors should consider the detention of genes and the simplicity and details of the description of methods for specific improvements regarding the methodology.
[Response]: We appreciate the reviewer’s insightful comment on improving the clarity and detail of our methodology, particularly regarding gene detection. To address this, we have revised the Genomic DNA extraction section to include additional information on the culture conditions, lysis procedure optimized for Gram-positive bacteria, and DNA quality assessment. These details will help enhance the reproducibility and transparency of our experimental workflow.
Please refer to Section 5.3 (pages 7–8, lines 267–277) in the revised manuscript for the updated description.
Reviewer 2 Report
Comments and Suggestions for Authors
This communication is a report about the identification of a Clostridium botulinum strain in Korea with both bont/B and bont/F gene clusters. The study aims to give genomic and functional characteristics of this dual-toxin-producing strain.
The topic is very interesting and it can be important for public health.
In my opinion the paper is well written. The introduction gives a proper background about C. botulinum infections, with particular attention to strains possessing dual-toxin gene clusters. The methods are properly described and the results are clearly presented and discussed.
Author Response
Response to Reviewer 2
We thank Reviewer 2 for the positive and constructive comments on our manuscript. We are pleased that the background, methods, and results were found to be clearly presented, and we appreciate your recognition of the relevance and clarity of our study.
Reviewer 3 Report
Comments and Suggestions for Authors
First of all, thank you for your contribution. The article has made certain new contributions to the identification of Clostridium botulinum strains capable of producing dual toxins. This is a very interesting topic, but I still have a few questions that need your explanation:
1.In the introduction section, you can supplement the comparison of the prevalence of dual-toxin-producing strains between South Korea and other countries to enhance the global perspective of the study.
2.Please explain why only three temperature points (30°C, 35°C, and 37°C) were tested. Is there no need to add more gradients to comprehensively analyze the impact range of temperature on toxin expression?
3.Can the positions of key regulatory genes be highlighted in Figure 1 to visually display the sequence similarity?
4.The article mentions the "B5" and "F2" subtypes. It is recommended to briefly explain the basis for subtype classification to avoid comprehension difficulties for non-specialist readers.
5.It is recommended that when abbreviations first appear, their full names should be supplemented, such as TPGY on line 76.
6.There are only 4 articles published after 2020 in the references. Is it possible to cite more literatures from the past five years to reflect the advancement of the research?
7.In the discussion section, you can supplement specific data on the differences in clinical symptoms and treatment difficulty between dual-toxin strains and single-toxin strains.
Author Response
Response to Reviewer 3
We sincerely appreciate your detailed and thoughtful comments. Your feedback helped us to refine our manuscript and clarify key points. We have carefully addressed each of your suggestions, and corresponding changes have been made in the revised manuscript.
[Comments 1]: In the introduction section, you can supplement the comparison of the prevalence of dual-toxin-producing strains between South Korea and other countries to enhance the global perspective of the study.
[Response 1]: By comparing the prevalence of dual-toxin-producing strains in South Korea and the United States, we emphasized that the identification of a strain producing both BoNT/B and BoNT/F in a foodborne botulism case in Korea is a globally rare finding. This additional content has been incorporated into the Introduction (page 2, lines 66–73).
[Comments 2]: Please explain why only three temperature points (30°C, 35°C, and 37°C) were tested. Is there no need to add more gradients to comprehensively analyze the impact range of temperature on toxin expression?
[Response 2]: In 2014, strain CB-2014001 was initially identified as producing only BoNT/B based on mouse bioassay results using cultures grown at 37°C. At that time, real-time PCR and whole-genome sequencing methods had not yet been actively implemented, and only BoNT/B expression was detected through a mouse bioassay. However, in 2020, both real-time PCR and genomic analysis revealed that this strain harbors bont/F in addition to bont/B. Based on prior research (Barash, J.R. and Arnon, S.S. 2004), we recognized that incubation temperature plays a critical role in distinguishing the expression profiles of BoNT/B and BoNT/F. Accordingly, we selected three biologically relevant temperatures—30°C, 35°C, and 37°C—for our analysis. These temperatures were sufficient to demonstrate differential expression of the two toxins and to address the study’s aims without the need for additional temperature gradients.
[Comments 3]: Can the positions of key regulatory genes be highlighted in Figure 1 to visually display the sequence similarity?
[Response 3]: We would like to clarify that all genes within the bont gene cluster are functionally important, which makes it difficult to specifically highlight only certain genes for special emphasis. Instead, to allow readers to easily compare sequence similarity across strains, we have included Average Nucleotide Identity (ANI) values for each gene and connected homologous genes with dotted lines in Figure 1. We believe this approach provides a clear and comprehensive visualization of gene conservation throughout the cluster, addressing the reviewer’s request for improved visual display of sequence similarity. Accordingly, we have revised Figure 1 and updated the corresponding legend and main text in the revised manuscript (page 3, lines 108–113).
[Comments 4]: The article mentions the "B5" and "F2" subtypes. It is recommended to briefly explain the basis for subtype classification to avoid comprehension difficulties for non-specialist readers.
[Response 4]: In response, we have added a brief explanation of the BoNT subtype classification criteria in the revised manuscript (page 3, lines 100–102). Specifically, subtypes such as B5 and F2 are distinguished based on amino acid sequence differences in the BoNT protein, with a divergence of more than 2.6% typically used to define separate subtypes within the same serotype.
[Comments 5]: It is recommended that when abbreviations first appear, their full names should be supplemented, such as TPGY on line 89.
[Response 5]: We have revised the manuscript to include the full name of Tryptone Peptone Glucose Yeast extract (TPGY) at its first mention on line 255.
[Comments 6]: There are only 4 articles published after 2020 in the references. Is it possible to cite more literatures from the past five years to reflect the advancement of the research?
[Response 6]: We have added recent references published after 2020 to reflect the latest developments in botulinum toxin diagnostics and therapeutics. Specifically, references [23] and [24] have been cited in the revised manuscript to strengthen the relevance and academic depth of the Discussion section (page 7, lines 220).
[Comments 7]: In the discussion section, you can supplement specific data on the differences in clinical symptoms and treatment difficulty between dual-toxin strains and single-toxin strains.
[Response 7]: The discussion referred to statistical observations comparing the clinical symptoms of dual-toxin-producing and single-toxin-producing strains, rather than results derived from controlled clinical trials (Panditrao et al. 2023). In our study, only one case of foodborne botulism involving a dual-toxin-producing strain has been reported in Korea, and the available clinical information is limited. Therefore, it is not sufficient to evaluate the severity differences between the two toxin types. Accordingly, we did not attempt to describe clinical differences in this manuscript, but we emphasized the need for future research based on more accumulated data.

Reviewer 4 Report
Comments and Suggestions for Authors
The manuscript provides important and relevant information. To better valorize the results I strongly suggest following improvement.
Major revisions needed:
State whether clusters reside on a plasmid or chromosome and supply complete genome/reads in GenBank. Recent B(F) pâté-outbreak genomes showed plasmid-borne B5+F2 cluster; direct comparison would strengthen novelty.
Toxin quantification: neutralisation at a single 1:640 dilution (n = 2 mice) cannot rank titres. Provide Endopep-MS, ELISA or LD50 curves and statistics to substantiate the “~4-fold” claim.
Add brief case synopsis (food vehicle, clinical course, treatment).
Examine botR/promoter variants that may drive temperature bias and relate to prior mechanistic work on dual-toxin strains.
Describe diagnostics outlook captures all recognized bont subtypes and outline how Korean surveillance could incorporate such assays.
Minor points
Update antitoxin paragraph: US BAT heptavalent antitoxin (FDA 2024) is effective against BoNT/F.
Author Response
Response to Reviewer 4
We thank you very much for your insightful comments and suggestions. Your input was instrumental in improving the clarity and scientific rigor of our manuscript. Below, we provide point-by-point responses to your concerns.
Major revisions needed:
[Comments 1]: State whether clusters reside on a plasmid or chromosome and supply complete genome/reads in GenBank. Recent B(F) pâté-outbreak genomes showed plasmid-borne B5+F2 cluster; direct comparison would strengthen novelty.
[Response 1]: We sincerely appreciate the reviewer’s insightful comments and suggestions. We fully acknowledge the importance of full genome disclosure and comparative analysis for scientific rigor. However, due to biosafety regulations and institutional policies within the Korea Disease Control and Prevention Agency (KDCA), we are unable to deposit the complete genome sequences of high-risk bacterial strains such as CB-2014001 in public databases. We kindly ask for your understanding regarding this limitation.
To address the reviewer’s request for comparison with recent plasmid-borne B5+F2 cluster genomes, we have added the PA9508B strain (GenBank accession number CP102918) to Figure 1. Furthermore, we have included the Average Nucleotide Identity (ANI) values for each gene compared to CB-2014001, and connected homologous genes across strains with dotted lines to visually highlight sequence conservation patterns.
Our analysis confirms that both the bont/B5 and bont/F2 gene clusters in CB-2014001 are located on a plasmid, consistent with the arrangement observed in the 1995 pâté-outbreak genomes described by PA9508B. Although we cannot provide the complete genome sequence, the bont gene cluster sequences (GenBank OP019001, OP019002) and raw sequencing data for these regions can be made available upon request and under appropriate institutional agreements.
We believe that these modifications and clarifications, along with the updated Figure 1 and Figure 2 (page 3–4), allow for meaningful comparative analysis while complying with regulatory requirements, and we hope this addresses the reviewer’s concerns regarding novelty and scientific rigor.
[Comments 2]: Toxin quantification: neutralisation at a single 1:640 dilution (n = 2 mice) cannot rank titres. Provide Endopep-MS, ELISA or LD50 curves and statistics to substantiate the “~4-fold” claim.
[Response 2]: In this study, we performed a mouse bioassay using strain CB-2014001, which was isolated in 2014, to confirm the expression of both BoNT/B and BoNT/F. The mouse bioassay remains the gold standard for botulism diagnosis and is used to determine toxin presence based on survival outcomes. The number of mice per condition (n = 2) was selected in accordance with WHO guidelines for botulism testing, which recommend minimal animal use under ethical principles when screening for toxin presence.
The assay design was based on the study by Barash and Arnon (Barash, J.R. and Arnon, S.S. 2004), which utilized dilution series to estimate lethal toxin concentrations in culture supernatants. In our study, we similarly employed serial dilutions to assess the relative toxicity of culture supernatants obtained at 30 °C and 37 °C. Although the original title of Table S2 was “Quantitative analysis of toxin production levels at 30 °C and 37 °C, based on mouse bioassay results,” the data do not allow for precise quantification or statistical ranking of toxin titers. Therefore, we revised the title of Table S2 to “Mouse lethality of CB-2014001 filtrates at 30 °C and 37 °C across dilution series” to more accurately reflect the nature of the data.
In line with this change, we also revised the corresponding sentence in the manuscript (page 6, lines 168–172) to clarify that the findings were based on lethal dilution rather than quantitative analysis:
[Comments 3]: Add brief case synopsis (food vehicle, clinical course, treatment).
[Response 3]: We agree that including a brief synopsis of the patient case would enhance the clinical context of our study. However, the 2014 foodborne botulism case in Korea involved a male adolescent who had consumed canned ham over an extended period. Due to ongoing legal proceedings involving the food manufacturer at the time, the patient's clinical details and treatment information were not disclosed, and no official case report has been published.
Therefore, we were unable to add further details regarding the clinical course or treatment. This limitation has now been briefly stated in the revised manuscript (page 2, lines 63–64) as follows: “However, no official case report detailing the clinical course or treatment has been published to date.”
[Comments 4]: Examine botR/promoter variants that may drive temperature bias and relate to prior mechanistic work on dual-toxin strains.
[Response 4]: We have addressed the reviewer’s suggestion regarding potential botR or promoter-based temperature bias in the revised manuscript (page 6, lines 179–192). Based on previous studies, we noted that temperature-dependent differences in BoNT expression are not likely caused by genetic variation in botR or promoter sequences, but rather by post-transcriptional regulation and protein stability. This discussion was incorporated to provide mechanistic insight into the observed temperature-specific expression of BoNT/B and BoNT/F.
[Comments 5]: Describe diagnostics outlook captures all recognized bont subtypes and outline how Korean surveillance could incorporate such assays.
[Response 5]: To reflect your suggestion, we have added a brief outlook on diagnostic strategies and Korean surveillance efforts in the revised manuscript (page 6–7, lines 207–228). In Korea, only 11 confirmed cases of botulism have been reported over the past 23 years, which remains relatively low even when adjusted for population size compared to countries such as the United States. Nevertheless, botulinum neurotoxin is classified as a Group 1 notifiable infectious disease and a potential bioterrorism agent under Korean public health regulations, warranting continuous vigilance and preparedness. Specifically, we noted that dual-toxin-producing strains have been identified in two separate cases in the past decade (in 2014 and 2019), including one case of infant botulism.
Although the exact mechanism behind the emergence of such strains remains unclear, it is plausible that ongoing microbial genetic variation—possibly as an adaptive response to global climate and environmental changes—may contribute to these events. In this context, strengthening surveillance systems to address the potential risk of future cases is essential.
Accordingly, Korea is actively working to improve its diagnostic capabilities, including the development of rapid detection kits and genome-based analytical tools for botulism. However, as detailed implementation plans are still under discussion, we are unable to include them formally in the manuscript at this time. We kindly ask for your understanding regarding the current limitations in reporting policy-level details.
Minor points
[Comments 6]: Update antitoxin paragraph: US BAT heptavalent antitoxin (FDA 2024) is effective against BoNT/F.
[Response 6]: In response, we have revised the antitoxin-related paragraph in the Discussion section (page 6-7, lines 216–217) to briefly include information on the botulism antitoxin heptavalent (BAT), which is effective against all seven BoNT serotypes (A–G), including BoNT/F. The updated sentence reflects the broader coverage of BAT and addresses the reviewer’s comment appropriately.

Round 2
Reviewer 3 Report
Comments and Suggestions for Authors
The article has been greatly improved, it is recommended to accept it
Reviewer 4 Report
Comments and Suggestions for Authors
The work was improved and in my opinion is suitable for publication.
Good luck.